# Comprehensive Evaluation and Construction of Drought Resistance Index System in Hulless Barley Seedlings

**DOI:** 10.3390/ijms26083799

**Published:** 2025-04-17

**Authors:** Liping Niu, La Bo, Shuaihao Chen, Zhongmengyi Qin, Dawa Dondup, Lhundrup Namgyal, Xiruo Quzong, Zhuo Ga, Yanming Zhang, Yafei Shi, Xin Hou

**Affiliations:** 1School of Ecology and Environment, Tibet University, Lhasa 850002, China; lipingniu@whu.edu.cn (L.N.); 13260090621@163.com (L.B.); csh102050@126.com (S.C.); Qinzmy4205@163.com (Z.Q.); sklhr@whu.edu.cn (Y.Z.); 2State Key Laboratory of Hulless Barley and Yak Germplasm Resources and Genetic Improvement, Lhasa 850002, China; dwdunzhu@126.com (D.D.); lhundrupnamgyal@163.com (L.N.); quzongxr160417@163.com (X.Q.); zhuogalahappy@126.com (Z.G.); 3College of Life Sciences, Wuhan University, Wuhan 430072, China; 4College of Life Sciences, Xinyang Normal University, Xinyang 464000, China; 5Wuhan University Shenzhen Research Institute, Shenzhen 518057, China

**Keywords:** drought tolerance, photosynthetic efficiency, oxidative stress, hulless barley, seedling-stage evaluation

## Abstract

With global climate change ongoing, the frequency and intensity of extreme weather events have increased annually. Hulless barley (*Hordeum vulgare* L. var. nudum), a primary crop cultivated in the Qinghai–Tibet Plateau mountains, frequently encounters multiple abiotic stresses including low temperature, high salinity, and drought. Among these stresses, drought has emerged as a critical environmental constraint affecting sustainable agricultural development worldwide. Establishing a drought resistance evaluation system for the hulless barley germplasm during its seedling stages could provide a theoretical foundation for screening and breeding drought-tolerant cultivars to address climate change challenges. This study employed two drought-sensitive (YC85 and YC88) and two drought-tolerant (ZY1252 and ZY1100) cultivars to develop an effective drought resistance evaluation protocol for hulless barley. Our findings identified several reliable indicators for assessing drought tolerance at the seedling stage: fresh mass, chlorophyll fluorescence parameters (F_v_/F_m_, NPQ, and *R*_FD_), photosynthetic parameters (E and gsw), and reactive oxygen species (ROS) levels. The established evaluation system was subsequently applied to three uncharacterized cultivars (ZY673, ZY1403, and KL14). The results classified all three as drought-sensitive, with ZY1403 exhibiting the highest sensitivity. Our work has established a comprehensive drought resistance evaluation framework for Tibetan hulless barley. Furthermore, this study provides valuable insights for optimizing cultivation practices and water resource management strategies, offering theoretical guidance for agricultural adaptation to climate change.

## 1. Introduction

Climate change-driven alterations in precipitation regimes, coupled with escalating water scarcity, are exerting unprecedented pressure on global agricultural systems, threatening food security for a rapidly expanding population projected to reach 9.7 billion by 2050 [1,2,3]. To meet this demand, current agricultural productivity must increase by 60–110%, yet recurrent droughts—intensified by anthropogenic warming—have severely constrained crop yields, destabilizing food supply chains worldwide [4,5,6]. Drought, as a multifaceted abiotic stressor, disrupts plant growth through mechanisms including cellular dehydration, oxidative damage, and metabolic dysregulation, ultimately impairing photosynthesis, nutrient assimilation, and reproductive success [7,8,9,10]. In response, plants deploy three evolutionary strategies: (i) drought escape, achieved by accelerating developmental phases to complete life cycles before stress onset; (ii) drought avoidance, involving enhanced water uptake (e.g., via deep root systems) or reduced transpiration (e.g., stomatal closure); and (iii) drought tolerance, which enables sustained physiological activity under a low water potential through osmotic adjustment and ROS scavenging [11,12,13,14,15]. Deciphering the molecular and physiological underpinnings of these strategies is critical for engineering climate-resilient crops and optimizing water-use efficiency in agroecosystems [16,17,18,19].

A central consequence of drought is the destabilization of photosynthetic machinery, characterized by reduced chlorophyll content, diminished stomatal conductance, and an impaired maximum quantum yield of PSII (F_v_/F_m_) [20,21]. Chlorophyll fluorescence parameters, particularly non-photochemical quenching (NPQ) and F_v_/F_m_, have emerged as robust biomarkers for drought tolerance screening, directly reflecting PSII’s repair capacity and thylakoid membrane stability under stress [22,23]. Mechanistically, drought-induced photoinhibition arises from the accumulation of undissipated excitation energy, which concurrently damages both Photosystem I (PSI) and PSII, leading to systemic photosynthetic collapse [24]. Divergent photoprotective strategies have evolved across species: In Arabidopsis, cryptochrome-deficient mutants (cry1cry2) exhibit enhanced drought tolerance, whereas CRY1 overexpression elevates transpiration via COP1-mediated stomatal regulation [25]. Wheat reconfigures its photosynthetic apparatus through the drought-responsive phosphorylation of PSII-LHCII complexes and dynamic adjustments in protein stoichiometry [26]. Monocots display further specialization—rice prioritizes NPQ activation via PsbS-mediated pH-dependent energy dissipation [27,28], while maize employs a dual strategy combining energy-dependent quenching (qE) and state transitions (qT), regulated by xanthophyll cycle dynamics and LHCII reorganization [29]. These adaptations are invariably accompanied by the drought-induced suppression of photosynthetic parameters (Pn, Gs, Ci) and chlorophyll degradation [30]. Despite these advances, research on hulless barley, a staple crop in arid, high-altitude regions, remains fragmented, with limited genomic insights into its drought-responsive traits.

To mitigate ROS-mediated damage under drought, plants deploy synergistic antioxidant systems comprising enzymatic scavengers (e.g., superoxide dismutase [SOD] and catalase [CAT]) and non-enzymatic mechanisms like the ascorbate–glutathione cycle [31,32]. Drought-triggered ROS overproduction—driven by NADPH oxidase activation and electron transport chain leakage—induces oxidative damage to lipids, proteins, and nucleic acids, necessitating the rapid upregulation of antioxidant defenses to restore redox balance [33,34,35]. Species-specific regulatory networks have evolved: rice enhances drought resilience through the coordinated induction of OsSOD and OsCAT [36], while maize optimizes stomatal conductance alongside ZmSOD/ZmCAT activation to balance carbon fixation and water conservation [37,38]. Hulless barley, endemic to the Qinghai–Tibet Plateau, exemplifies unique adaptations to extreme aridity, erratic precipitation, and intense UV-B radiation [39,40]. Genomic analyses reveal drought-responsive alleles (HvDREB1, HvNAC72) that orchestrate osmolyte biosynthesis and cell wall remodeling, while ROS management is fine-tuned via HvbZIP21-mediated scavenging and HvAKT1-regulated ion homeostasis [41,42]. Notably, *HvMORF8* mitochondrial gene silencing disrupts antioxidant coordination, exacerbating ROS accumulation and hydraulic dysfunction under drought [43]. These findings underscore hulless barley’s evolutionary innovations—integrating ROS control, epigenetic regulation, and root–shoot signaling—that remain underexploited for improving stress resilience in temperate cereals.

Recent efforts have begun mapping drought tolerance traits across hulless barley’s growth stages [44,45,46], yet a systematic framework for evaluating its drought resistance mechanisms remains elusive. This study addresses this gap by establishing a comprehensive drought resistance evaluation system for the hulless barley germplasm. We integrate multi-omics data and physiological profiling to identify key drought tolerance markers. Our research aims to uncover the genetic and biochemical mechanisms behind hulless barley’s drought resistance, develop effective screening methods for drought-tolerant varieties, and advance sustainable breeding and water management strategies for arid-region cultivation.

## 2. Results

### 2.1. PEG Treatment Simulates Drought Stress in Hulless Barley Cultivars

Cultivating drought-resistant cultivars of hulless barley is the most effective and economical strategy to alleviate the adverse impacts of drought stress on hulless barley production [47]. However, the genetic diversity of cultivars of hulless barley has significantly decreased over the course of long-term domestication, particularly in the context of modern breeding and cultivation practices. Additionally, the current assessment of drought resistance in hulless barley germplasm resources lacks sufficient depth. Therefore, it is imperative to undertake a drought resistance identification and screening of the extensive hulless barley germplasm resources. Polyethylene glycol (PEG), as an osmotic regulator, is capable of effectively simulating different levels of drought stress [48]. To establish an efficient drought resistance evaluation system for hulless barley, we selected four cultivars from hulless barley germplasm resources (Appendix A): two drought-sensitive cultivars (YC85 and YC88) and two drought-tolerant cultivars (ZY1252 and ZY1100). After exposing different hulless barley cultivars to simulated drought conditions using 16% PEG for 2 days, we observed phenotypic differences between the drought-tolerant cultivars and drought-sensitive cultivars (Figure 1A). We found that the shoot lengths of the four hulless barley cultivars were almost the same (Figure 1B). The drought-sensitive cultivars (YC85 and YC88) showed a reduced fresh and dry mass with the 16% PEG treatment compared to without the PEG treatment. In contrast, for the drought-tolerant cultivars (ZY1252 and ZY1100), there were no significant differences in fresh and dry mass between the PEG-treated and untreated cases (Figure 1C,D). But the chlorophyll contents remained unchanged in the four hulless barley cultivars (Figure 1E–G). So, PEG had a negative effect on both the drought-sensitive hulless barley cultivars and their fresh mass, which can serve as a drought resistance index due to its convenience for statistical analysis.

### 2.2. PEG-Simulated Drought Treatment Had a Significant Effect on Photosynthetic Parameters

Photosynthesis in plants is highly sensitive to drought stress, and its impact is primarily reflected through parameters such as ‘A’, ‘E’, and stomatal conductance [49]. We observed that in drought-sensitive cultivars (YC85, YC88), the photosynthetic parameters decreased significantly under PEG treatment. However, in drought-resistant cultivars (ZY1252, ZY1100), most of these parameters did not undergo significant changes under PEG treatment (Figure 2). Therefore, we selected ‘E’ and ‘gsw’ as drought resistance indices due to their large range of variation and stability.

Previous studies have shown that the photosynthetic parameters in drought-sensitive cultivars are significantly affected by drought stress, with PSII parameters being the most strongly impacted [50]. Consequently, we measured the chlorophyll fluorescence parameters in various hulless barley cultivars to further evaluate the differences among them. The results indicated that in the drought-tolerant cultivars (ZY1100 and ZY1252), none of the chlorophyll fluorescence parameters underwent significant changes under PEG treatment (Figure 3). Conversely, in the drought-sensitive cultivars (YC85 and YC88), several parameters including F_v_/F_m_, NPQ, and the fluorescence decrease ratio (*R*_FD_) decreased significantly under PEG treatment. More specifically, F_v_/F_m_ decreased by 21% in YC85 and by 22% in YC88; NPQ decreased by 62% in YC85 and by 60% in YC88; and *R*_FD_ decreased by 60% in YC85 and by 50% in YC88. These findings suggest that F_v_/F_m_, NPQ, and *R*_FD_ are reliable indicators of simulated drought stress in hulless barley treated with PEG (Figure 3). The above results suggest that the photosynthetic parameters NPQ and *R*_FD_ can serve as indicators for drought resistance screening due to their large range of variation and stability.

### 2.3. PEG-Simulated Drought Treatment on Reactive Oxygen Species (ROS) Levels

Previous studies have shown that ROS accumulation is a defensive response to abiotic stress [51]. We measured these parameters in drought-sensitive cultivars (YC85 and YC88) and drought-tolerant cultivars (ZY1100 and ZY1252) of hulless barley after treating them with PEG for 2 days. The level of H_2_O_2_ in the drought-sensitive cultivars (YC85 and YC88) increased dramatically under PEG treatment, whereas it remained unchanged in the drought-tolerant cultivars (ZY1100 and ZY1252). Interestingly, the content of O^2−^ did not exhibit significant changes under PEG treatment in any of the four cultivars. These results indicate that H_2_O_2_ content is a more reliable indicator of simulated drought stress than O^2−^ content in hulless barley (Figure 4). Similarly, the staining intensity of DAB can serve as an indicator for drought resistance screening due to the huge differences between drought-sensitive and drought-tolerant cultivars.

### 2.4. Verification of Drought Resistance Evaluation System

To validate the drought resistance evaluation system in hulless barley, we analyzed the fresh mass, photosynthetic parameters (‘E’, ‘gsw’), chlorophyll fluorescence (NPQ, *R*_FD_), and DAB-stained H₂O₂ levels in seedlings. Five cultivars were selected: drought-tolerant ZY97, drought-sensitive YC83, and three uncharacterized lines (ZY673, ZY1403, KL14). All cultivars were subjected to PEG-induced drought stress and control conditions (CK). Comparative analysis revealed that the three uncharacterized cultivars displayed drought-sensitive responses under PEG treatment, mirroring the sensitive control. Notably, the PEG-treated plants showed significant reductions: their fresh mass was 39%, 30%, and 34% lower than the CK treatment (Figure 5A); the ‘E’ declined by 89%, 96%, and 50% (Figure 5B); and the ‘gsw’ declined by 95%, 97%, and 53% of the controls (Figure 5C). The chlorophyll fluorescence parameters exhibited similar trends: NPQ declined by 54%, 75%, and 35% (Figure 5D), while *R*_FD_ declined by 52%, 68%, and 36% (Figure 5E). ZY1403 demonstrated the highest sensitivity, followed by ZY673 and KL14. DAB staining confirmed elevated H₂O₂ accumulation under drought, particularly in ZY1403 (Figure 5F). These results collectively validate our evaluation system, conclusively classifying all three uncharacterized cultivars as drought-sensitive, with ZY1403 showing the most pronounced sensitivity.

## 3. Discussion

Drought stress imposes multifaceted constraints on plant growth by disrupting water relations, photosynthetic efficiency, and redox homeostasis [7,8]. This study systematically evaluated the physiological divergence between drought-tolerant and drought-sensitive hulless barley cultivars under PEG-simulated drought stress, establishing a comprehensive evaluation framework integrating morphological, photosynthetic, and oxidative stress biomarkers. Our findings not only advance the understanding of hulless barley’s unique adaptations to arid, high-altitude environments but also provide actionable insights for breeding climate-resilient crops.

Consistent with observations in wheat and rice under water deficit [52,53,54,55], the drought-sensitive hulless barley cultivars exhibited significant reductions in fresh mass (Figure 1C), ‘E’ (Figure 2B), and ‘gsw’ (Figure 2C), whereas the tolerant cultivars maintained a stable biomass and photosynthetic activity. This divergence aligns with the “drought avoidance” strategy, where tolerant genotypes prioritize resource allocation to sustain growth under stress [11,56]. Notably, the stability of the chlorophyll a and b content across all cultivars (Figure 1E–G) contrasts with reports in maize and peanuts, where drought-induced ROS accumulation accelerated chlorophyll degradation [57,58]. We propose that hulless barley’s compact leaf morphology—characterized by a reduced leaf area and thickened cuticles—may mitigate oxidative damage by lowering light absorption and ROS penetration, a trait likely honed by its adaptation to high-altitude UV-B radiation [59].

Photosynthetic parameters, particularly F_v_/F_m_, NPQ, and *R*_FD_, emerged as robust proxies for drought sensitivity. The tolerant cultivars maintained their F_v_/F_m_ and sustained their NPQ under drought (Figure 3), indicative of enhanced photoprotective mechanisms. Our findings align with the drought-induced responses observed in Atractylodes lancea, characterized by suppressed photosynthetic parameters and diminished chlorophyll fluorescence, as further corroborated by RNA-seq analysis [30]. The stability of NPQ in the tolerant cultivars suggests the efficient thermal dissipation of excess energy, a critical adaptation in high-radiation environments [28,60]. Conversely, the sensitive cultivars exhibited sharp declines in F_v_/F_m_, NPQ, and *R*_FD_ (Figure 3D–F), reflecting impaired PSII repair and thylakoid membrane destabilization. These findings align with reports in mung bean, where sustained NPQ correlated with drought tolerance through upregulated photoprotective proteins [61,62,63]. Furthermore, the stability of ‘E’ and ‘gsw’ in the tolerant cultivars (Figure 2) underscores their ability to balance water conservation with carbon assimilation, a hallmark of drought resilience [64].

Differential ROS regulation further distinguished the tolerant and sensitive cultivars. While H_2_O_2_ levels surged in the sensitive cultivars under drought (Figure 4A), the tolerant lines maintained redox equilibrium, likely through enhanced enzymatic scavenging (e.g., SOD, CAT) or non-enzymatic antioxidants (e.g., ascorbate–glutathione cycle) [65,66]. This aligns with proteomic studies in barley, where drought-tolerant cultivars exhibited upregulated ROS detoxification pathways [33,67]. Notably, O^2−^ levels remained unchanged across the cultivars (Figure 4B), suggesting rapid dismutation to H_2_O_2_ via SOD activity—a mechanism observed in *Arabidopsis* under osmotic stress [68,69]. The strong correlation between DAB staining intensity and H_2_O_2_ accumulation (Figure 5F) reinforces H_2_O_2_ as a reliable oxidative stress marker for high-throughput drought screening.

The evaluation system, validated using uncharacterized cultivars (ZY673, ZY1403, KL14; Figure 5), successfully classified genotypes based on fresh mass, photosynthetic parameters, and ROS dynamics. For instance, ZY1403—the most sensitive cultivar—exhibited the steepest declines in ‘E’, ‘gsw’, and NPQ, coupled with pronounced H_2_O_2_ accumulation (Figure 5B–F). This multi-tiered approach addresses the limitations of traditional yield-based assessments by providing rapid, cost-effective screening tools for the early seedling stages [70,71]. Moreover, the integration of chlorophyll fluorescence (NPQ, *R*_FD_) and oxidative markers (DAB staining) offers a scalable framework for breeding programs targeting marginal agroecosystems.

While this study provides critical insights, two limitations warrant attention. First, the exclusion of root architectural traits and osmoprotectant dynamics (e.g., proline, soluble sugars) limits a holistic understanding of drought adaptation. Root depth and hydraulic conductivity are pivotal for water acquisition in field conditions [16], and their integration could enhance predictive accuracy. Second, acute PEG-induced drought may not replicate field-acclimated responses to cyclic stress. Future studies should combine controlled-environment assays with field trials under chronic drought, coupled with multi-omics approaches (e.g., transcriptomics, proteomics) to resolve the molecular networks underlying hulless barley’s resilience. For instance, profiling *HvDREB1* and *HvNAC72* expression—genes linked to drought tolerance in hulless barley [39,40]—could elucidate the regulatory mechanisms driving their observed physiological traits. Notably, our previous study revealed that 59 drought-responsive genes were differentially expressed between drought-sensitive (YC85) and drought-tolerant (ZY1100) varieties, as identified through comparative RNA-seq analysis [72].

By synthesizing morphological, photosynthetic, and oxidative stress metrics, this study establishes a robust framework for drought tolerance screening in hulless barley. The stability of the fresh mass, chlorophyll fluorescence parameters (F_v_/F_m_, NPQ, and *R*_FD_), photosynthetic parameters (‘E’ and ‘gsw’), and ROS levels in the tolerant cultivars highlights their utility as biomarkers for breeding programs. These findings not only elucidate hulless barley’s unique adaptations to arid, high-altitude ecosystems but also pave the way for developing climate-resilient barley germplasms through targeted trait introgression. This study has established a streamlined, high-throughput phenotypic evaluation system for drought resistance in hulless barley. The developed platform provides an efficient tool for the large-scale screening of drought-tolerant genotypes and facilitates molecular breeding programs targeting water-use efficiency. Our results demonstrate strong positive correlations between chlorophyll fluorescence parameters (F_v_/F_m_, NPQ, and *R*_FD_) and biomass accumulation under progressive drought stress. Furthermore, integrating phenomics-assisted selection with IoT-enabled field monitoring technologies enables revolutionary advancements in drought phenotyping. Specifically, UAV-mounted chlorophyll fluorescence imaging systems coupled with multispectral sensors achieve accurate predictions of drought tolerance indices. This integrated approach significantly enhances field-based drought assessment throughput while providing actionable insights for precision irrigation scheduling and cultivar-specific agronomic management in hulless barley production systems.

## 4. Materials and Methods

### 4.1. Plant Materials

We selected the drought-tolerant cultivars ZY1100, ZY1252, and ZY97, as well as the drought-sensitive cultivars YC85, YC88, and YC83. Furthermore, due to their heightened susceptibility to environmental influences, YC85 and YC88 demonstrated a higher requirement for water and nutrient inputs. Additionally, we included cultivars with unknown drought resistance grades, namely ZY673, ZY1403, and KL14.

The drought-resistant and drought-sensitive hulless barley cultivars were selected based on established cultivation practices in the Qinghai–Tibet Plateau combined with preliminary experimental data accumulated by our laboratory. These selection criteria were derived from systematic observations of crop performance under controlled drought simulations conducted during our previous research cycles.

### 4.2. Plant Growth and Drought Treatments

The hulless barley seeds were imbibed in distilled water for 2 days and subsequently germinated on moist filter paper placed in 100 mm Petri dishes for an 8-day period. Following germination, the seedlings were transferred to a hydroponic system containing Yoshida nutrient solution for a 4-day growth period. At the one-leaf and one-tiller stage (13 days after germination), drought stress was simulated by treating the seedlings with a 16% (*w*/*v*) PEG solution (prepared in Yoshida nutrient solution, pH 5.7–5.8) as the drought treatment group (Drought). Seedlings grown in Yoshida nutrient solution (pH 5.7–5.8) without PEG served as the control group (Control). The nutrient solution was replaced every 2 days. The cultivation environment was maintained at 22 °C with a light intensity of 66 μmol·m^−2^·s^−1^ and a photoperiod of 16 h light/8 h darkness.

### 4.3. Morphological and Biomass Measurement

Phenotypic diversity was observed among the various cultivars after 2 days of PEG treatment. After 7 days of PEG treatment, the following traits were quantified for each cultivar: seedling height, aboveground fresh mass, and dry mass (after desiccation at 65 °C for 48 h). A representative subset of 6 seedlings per cultivar was sampled. The experimental protocol was conducted in triplicate to ensure reproducibility.

### 4.4. Assessment of Photosynthetic Parameters

Gas exchange measurements were conducted from 09:00 a.m. to 4:00 p.m. on the hulless barley leaves after 2 days of 16% PEG treatment using a portable photosynthesis system coupled to a 3 × 3 cm^2^ leaf chamber (Li-6800; LICOR, Inc., Lincoln, NE, USA). The light intensity inside the leaf chamber was set to 100 μmol·m^−2^·s^−1^, and the CO_2_ concentration surrounding the leaf was maintained at 400 µmol mol^−1^ with a CO_2_ mixer. The relative humidity (60%) and leaf temperature (22 °C) were measured within an environment-controlled greenhouse. At the one-leaf and one-tiller stage (13 days after germination), following a 2-day exposure to 16% PEG-induced drought stress, the net photosynthetic rate (A), transpiration rate (E), stomatal conductance of water vapor (gsw), total conductivity to water vapor (gtw), intercellular CO_2_ concentration (Ci), and stomatal conductance of CO_2_ (gtc) were measured in the fully expanded first leaves of various plant cultivars, which were randomly sampled for analysis.

### 4.5. Measurement of Chlorophyll Fluorescence Parameters

The chlorophyll fluorescence parameters were quantified utilizing a FluorCam 800-C (PSI, Brno, Czech) chlorophyll fluorescence imaging system, as described [73]. After 2 days of 16% PEG-induced drought treatment, the fully expanded first leaves from different plants were randomly selected, and their chlorophyll fluorescence parameters were measured after 15 min in the dark.

### 4.6. Measurement of Chlorophyll Content

The leaves exposed to PEG and the control leaves were harvested and weighed, then soaked in 80% acetone containing 1 mg/mL of acetone and incubated in an oven for 16 h at 37 °C. The absorption values at 663 nm and 645 nm were measured using a NanoDrop 2000C (Thermo Scientific, Waltham, MA, USA) spectrophotometer. The concentrations of chlorophyll a (Chla) and chlorophyll b (Chlb) were calculated using the following formulas, as described [73]: chlorophyll a (mg/g FW) = (12.7 × A663) − (2.69 × A645); chlorophyll b (mg/g FW) = (22.9 × A645) − (4.68 × A663); and total chlorophyll content (mg/g FW) = chlorophyll a + chlorophyll b.

### 4.7. Detection of Reactive Oxygen Species (ROS)

Reactive oxygen species (ROS) were detected using the diaminobenzidine (DAB) and tetrazolium blue (NBT) staining methods, as described [74]. Solutions of 1 mg/mL DAB and 1 mg/mL NBT were prepared for the staining process. The first fully expanded leaf from the hulless barley seedlings that had been treated with 16% PEG for 3 days was immersed in the DAB and NBT staining solutions. Subsequently, the leaf was placed in a vacuum oven for 20–30 min, kept in the dark at room temperature for 2 days, and then bleached with 80% ethanol at 85 °C in a water bath prior to observation and photography. The staining intensity of the leaves was quantified using ImageJ (V1.53) (https://imagej.net/ij/ (accessed on 14 April 2025)).

### 4.8. Statistical Analysis

Origin 2022 software was used for all statistical tests. An unpaired Student’s *t*-test (2-tailed) was performed for the statistical analyses.

## Figures and Tables

**Figure 1 ijms-26-03799-f001:**
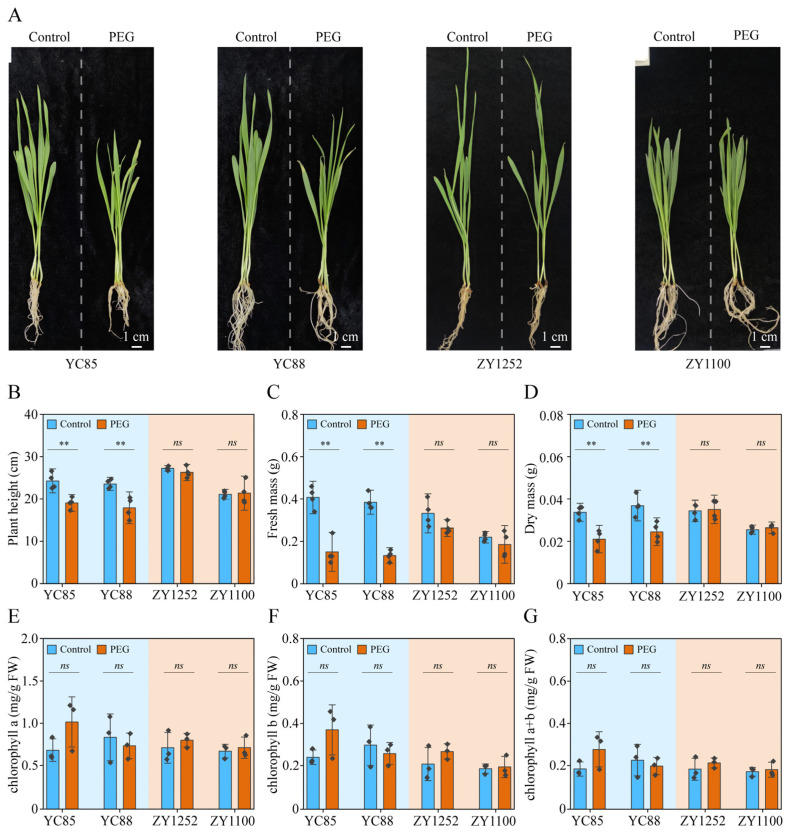
Effects of drought stress simulated by polyethylene glycol (PEG) on different cultivars of hulless barley plants. Drought-sensitive (YC85 and YC88) and drought-tolerant hulless barley (ZY1252 and ZY1100) varieties were selected for the experiment. (**A**) Phenotypes of hulless barley plants treated with or without PEG. Thirteen-day-old plants grown under Yoshida (Control) were treated with Yoshida or 16% PEG for another seven days. Bar = 1 cm. (**B**–**D**) Effect of PEG on shoot length (**B**), shoot fresh mass (**C**), and shoot dry mass (**D**) of hulless barley plants. Thirteen-day-old plants grown under Yoshida (Control) were treated with Yoshida or 16% PEG (Drought) for another seven days. The values indicate the means ± SDs (n = 4). ** *p* < 0.01 and ns *p* > 0.05, Student’s *t* test. (**E**–**G**), Effect of PEG on chlorophyll a (**E**), chlorophyll b (**F**), and chlorophyll a + b (**G**) of hulless barley plants. Thirteen-day-old plants grown under Yoshida (Control) were treated with Yoshida or 16% PEG for another three days. The values indicate the means ± SDs (n = 3). ** *p* < 0.01 and ns *p* > 0.05, Student’s *t* test.

**Figure 2 ijms-26-03799-f002:**
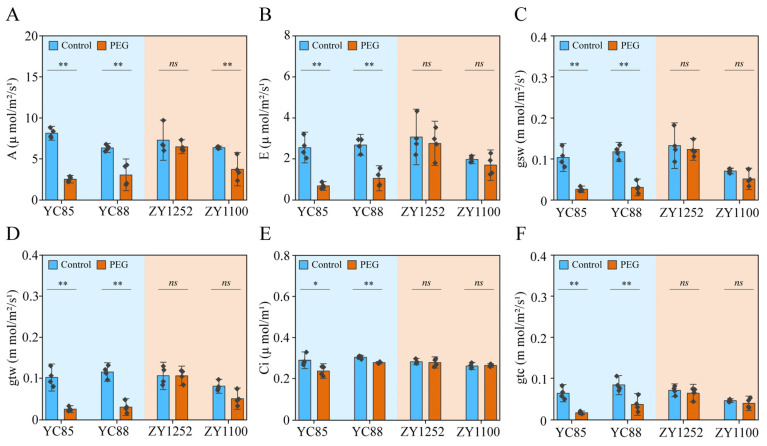
Photosynthetic physiological parameters of hulless barley plants under control conditions and PEG treatment. The photosynthetic gas exchange of YC85, YC88, ZY1252, and ZY1100 was measured with LI-6800 (LICOR, Inc., Lincoln, NE, USA) under control conditions and PEG treatment. Thirteen-day-old plants grown under Yoshida (Control) were treated with Yoshida or 16% PEG for another two days. Drought-sensitive (YC85 and YC88) and drought-tolerant hulless barley (ZY1252 and ZY1100) varieties were selected for the experiment. (**A**) Net photosynthetic rate (A); (**B**) transpiration rate (E). (**C**) Stomatal conductance of water vapor (gsw); (**D**) total conductivity to water vapor (gtw); (**E**) intercellular CO_2_ concentration (Ci), and (**F**) stomatal conductance of CO_2_ (gtc). The values indicate the means ± SDs (n = 4). ** *p* < 0.01, * *p* < 0.05 and ns *p* > 0.05, Student’s *t* test.

**Figure 3 ijms-26-03799-f003:**
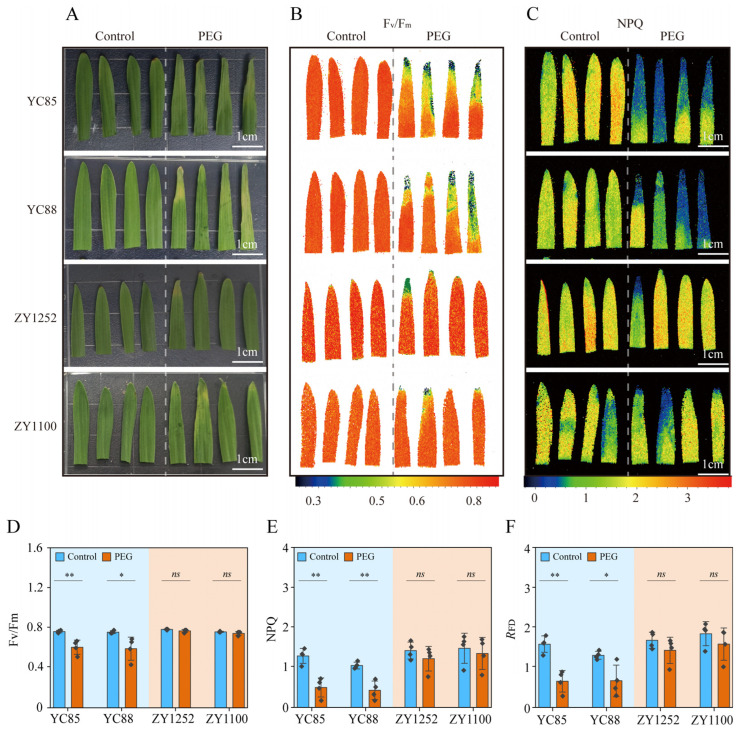
Chlorophyll fluorescence parameters of hulless barley plants under control conditions and PEG treatment. The photosynthetic gas exchange of the plants was measured with FluorCam (PSI, Brno, Czech Republic) under control conditions and PEG treatment. Thirteen-day-old plants grown under Yoshida (Control) were treated with Yoshida or 16% PEG for another two days. Drought-sensitive (YC85 and YC88) and drought-tolerant hulless barley (ZY1252 and ZY1100) varieties were selected for the experiment. The experimental materials were obtained from two different media (control and 16% PEG treatment), and the first fully unfolded leaf was taken from each medium after 15 min of dark adaptation for the measurement of their chlorophyll fluorescence parameters. (**A**) Phenotypes of hulless barley plants treated with or without PEG. Bar = 1 cm; (**B**) imaging of maximum quantum yield of PSII (F_v_/F_m_); (**C**) imaging of non-photochemical quenching (NPQ); (**D**) F_v_/F_m_; (**E**) NPQ; and (**F**) fluorescence decrease ratio (*R*_FD_). The values indicate the means ± SDs (n = 4). ** *p* < 0.01, * *p* < 0.05 and ns *p* > 0.05, Student’s *t* test.

**Figure 4 ijms-26-03799-f004:**
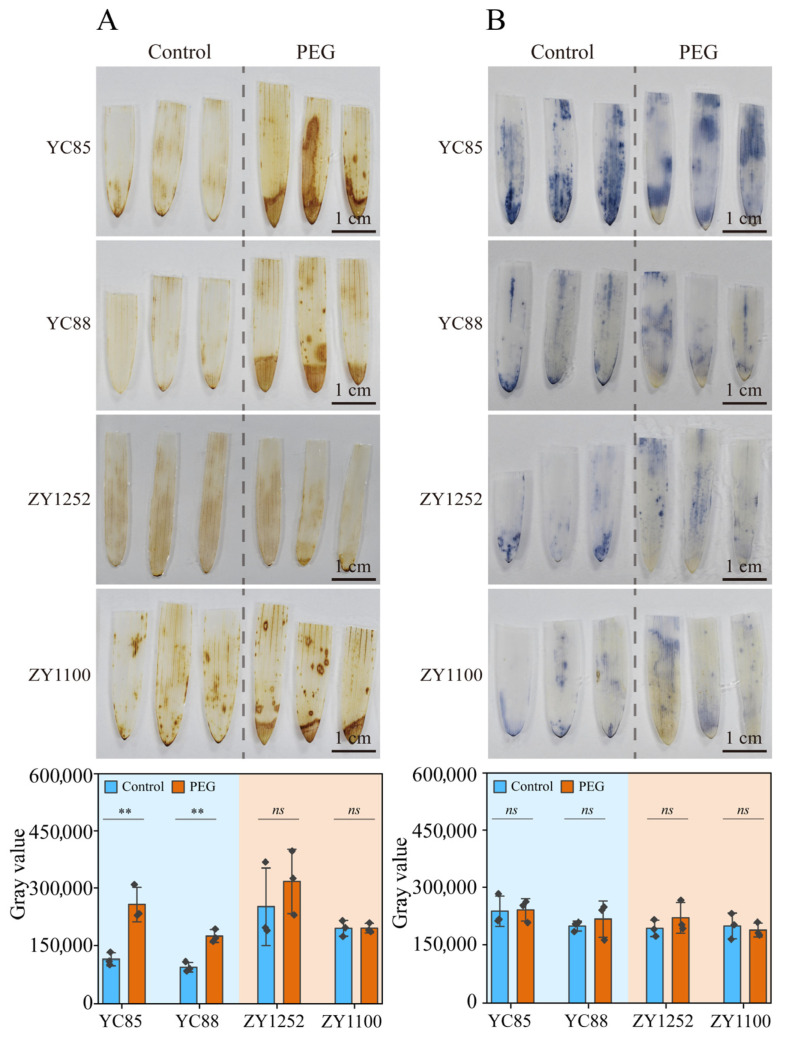
ROS accumulation in hulless barley plants under control conditions and PEG treatment. Thirteen-day-old plants grown under Yoshida (Control) were treated with Yoshida or 16% PEG for another three days. Drought-sensitive (YC85 and YC88) and drought-tolerant hulless barley (ZY1252 and ZY1100) varieties were selected for the experiment. (**A**) DAB staining was used to determine the hydrogen peroxide content under control conditions and PEG treatment, upper. The relative amount of DAB was estimated using ImageJV1.53, with measurements taken from the above plants, lower. (**B**) NBT staining was used to assess the superoxide anion in the control and under PEG treatment, upper. The relative amount of NBT was estimated using ImageJV1.53, with measurements taken from the above plants, lower. The values indicate the means ± SDs (n = 3). ** *p* < 0.01 and ns *p* > 0.05, Student’s *t* test.

**Figure 5 ijms-26-03799-f005:**
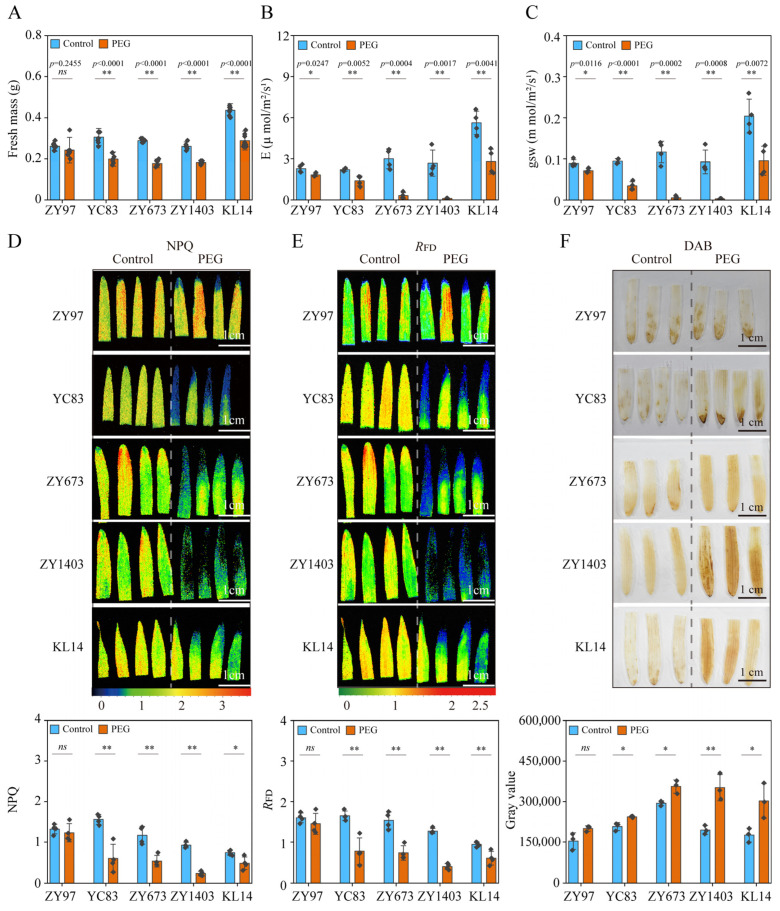
Verification of drought resistance evaluation. Thirteen-day-old seedlings of ZY97, YC83, ZY673, ZY1403, and KL14 were treated with 16% PEG in the drought group and were treated with Yoshida in the control group. The experimental materials were obtained from two different media, and the first leaf was taken from each medium for the measurement of different parameters. (**A**–**C**) Effect of PEG-simulated drought treatment on fresh mass (**A**), transpiration rate, ‘E’ (**B**), and stomatal conductance of water vapor, ‘gsw’ (**C**). The values indicate the means ± SDs (n = 8 for fresh mass; n = 4 for ‘E’ and ‘gsw’). ** *p* < 0.01 and * *p* < 0.05, Student’s *t* test. (**D**) The photosynthetic gas exchange of the plants was measured with FluorCam (PSI, Brno, Czech Republic) in the control and under PEG treatment. Imaging of non-photochemical quenching (NPQ), upper. The level of NPQ, lower. The values indicate the means ± SDs. ** *p* < 0.01, * *p* < 0.05 and ns *p* > 0.05, Student’s *t* test. (**E**) Imaging of the fluorescence decrease ratio (*R*_FD_), upper. The level of *R*_FD_, lower. The values indicate the means ± SDs (n = 4). ** *p* < 0.01 and ns *p* > 0.05, Student’s *t* test. (**F**) DAB staining was used to determine the hydrogen peroxide content in the control and under PEG treatment, upper. The relative amount of DAB was estimated using ImageJ (V1.53), with measurements taken from the above plants, lower. The values indicate the means ± SDs (n = 3). ** *p* < 0.01, * *p* < 0.05 and ns *p* > 0.05, Student’s *t* test.

## Data Availability

The data presented in this study are available in this article.

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
