# Peer review of "Comprehensive Evaluation and Construction of Drought Resistance Index System in Hulless Barley Seedlings"

_ijms, 2025, doi:10.3390/ijms26083799_

Round 1
Reviewer 1 Report
Comments and Suggestions for Authors
The manuscript's introduction is weak, with several instances where paragraphs lack logical connections (e.g., lines 52-58). Additionally, numerous paragraphs are missing references, including in the materials and methods and discussion sections.
Regarding the methodology, there are inconsistencies in the description of the varieties. The authors mention (lines 300-304) three drought-sensitive, three drought-tolerant, and three unknown varieties, yet in the results (lines 93-94), they refer to only two varieties, which do not match the descriptions in the materials and methods section or the figures.
Several crucial methodological details are missing:
- What were the seeds soaked in—distilled water? Some fungicide treatment?
- Were the petri dishes placed in a growth chamber? If so, what was the temperature and humidity?
- What was the nutrient concentration in the Yoshida solution? Was it diluted to reduce ionic strength and prevent osmotic shock?
- Was the PEG solution renewed during the 13-day treatment period? What were the pH and electrical conductivity of the solution at the beginning and end of the experiment?
- Why was the duration of water stress set at 13 days? What parameter or reference was used to determine this period? The description of this timeframe is unclear, as the methodology (lines 310-311) states one thing (13 days under PEG solution), while the figure captions (lines 144-145; 178; 208) indicate another (2 or 3 days). This creates confusion in understanding what was actually done.
- Were gas exchange measurements taken two days after the start of the water deficit? If so, why? Were any measurements conducted on the last day of water deficit? What light intensity was used in the LICOR during measurements, and at what time were they performed? Additionally, the authors mention “additional photosynthetic parameters were measured” (line 324) but do not specify which parameters or methods were used. The methodology section is to describe everything that was done and how.
- How long were the plants subjected to dark adaptation before chlorophyll fluorescence analysis?
- What references were used for chlorophyll concentration and ROS analyses?
- The data analysis section is missing. It is necessary to specify which statistical analyses were performed, which tests were applied, and which indices were used to determine significant differences. What was the experimental design?
The results indicate that drought-tolerant cultivars maintain physiological performance similar to control plants under stress, whereas drought-sensitive cultivars exhibit reduced performance. However, this is expected and does not provide novel insights. The discussion is weak and lacks integration with the findings. For example, why do the authors emphasize parameters such as gtw and gtc but fail to discuss them further? Additionally, no leaf water status parameters (e.g., relative water content, water potential) or other relevant metrics such as water use efficiency were measured, and in drought studies this data is of great importance. The reason for the lower NPQ in stressed plants is also not addressed.
Furthermore, the correct terminology is "fresh mass" and "dry mass," not "weight." Lastly, the measurement units of the analyzed parameters need to be reviewed.
Author Response
Comments and Suggestions for Authors
The manuscript's introduction is weak, with several instances where paragraphs lack logical connections (e.g., lines 52-58). Additionally, numerous paragraphs are missing references, including in the materials and methods and discussion sections.
Response:
Thank you for your suggestion. We have revised this accordingly.Line37-83.
Regarding the methodology, there are inconsistencies in the description of the varieties. The authors mention (lines 300-304) three drought-sensitive, three drought-tolerant, and three unknown varieties, yet in the results (lines 93-94), they refer to only two varieties, which do not match the descriptions in the materials and methods section or the figures.
Response:
We sincerely regret any ambiguity in our original description. For full transparency: (1) All barley germplasm accessions employed in this study are comprehensively documented in Lines 300-304; (2) The foundational materials for establishing the drought resistance evaluation framework comprised two drought-sensitive cultivars (YC85, YC88) and two drought-tolerant cultivars (ZY1252, ZY1100), as specified in Lines 93-94 and illustrated in Figures 1-4; (3) Validation trials subsequently implemented the drought-tolerant cultivar ZY97, drought-sensitive cultivar YC83, and three previously uncharacterized accessions (ZY673, ZY1403, KL14) to verify the system's discriminative accuracy.
Several crucial methodological details are missing:
- What were the seeds soaked in—distilled water? Some fungicide treatment?
Response:
Thank you for your valuable suggestions. In this experiment, the seeds were soaked in distilled water. The purpose of soaking was to ensure the seeds reached their saturation moisture content, which is necessary to trigger rapid and uniform germination. Distilled water was used to eliminate potential confounding factors from impurities or dissolved minerals, thereby standardizing the hydration process and ensuring consistent germination conditions across all experimental groups. Line312.
- Were the petri dishes placed in a growth chamber? If so, what was the temperature and humidity?
Response:
Thank you for your valuable suggestions. Yes, the petri dishes in the experiment were placed in a growth chamber. The temperature regime was set to a 16-hour light period at 22°C and an 8-hour dark period at 20°C, with humidity maintained at 60%. These details have now been explicitly incorporated into the revised manuscript to ensure completeness and clarity. Line333-335.
- What was the nutrient concentration in the Yoshida solution? Was it diluted to reduce ionic strength and prevent osmotic shock?
Response:
When the roots of each species grew to 2–3 cm, the distilled water was replaced with 1/2 Yoshida nutrient salt mixture (pH 5.7-5.8, Coolaber, Beijing, China, no. NSP1040) in a growth chamber. Line39-320.
- Was the PEG solution renewed during the 13-day treatment period? What were the pH and electrical conductivity of the solution at the beginning and end of the experiment?
Response:
In this study, the 13-day period refers to normal cultivation prior to PEG treatment. During the cultivation process, the Yoshida culture solution was renewed every 2 days, with the pH of the nutrient solution maintained at approximately 5.7-5.8. As the culture solution was regularly refreshed, we consider that the electrical conductivity of the solution remained stable and was not significantly affected throughout the experimental period.Line315-319.
- Why was the duration of water stress set at 13 days? What parameter or reference was used to determine this period? The description of this timeframe is unclear, as the methodology (lines 310-311) states one thing (13 days under PEG solution), while the figure captions (lines 144-145; 178; 208) indicate another (2 or 3 days). This creates confusion in understanding what was actually done.
Response:
Under controlled laboratory conditions, the barley seedlings required exactly 13 days to reach the one-leaf-one-heart developmental stage, as carefully validated through repeated growth trials in our controlled environment facility. Accordingly, the water stress treatment was administered precisely at this critical growth phase.
The discrepancies detected in lines 310-311 have been rectified through comprehensive revisions to ensure compliance with established documentation standards. The hulless barley seeds were selected and soaked for two days until they germinated and showed radicle emergence. The germinated seeds were then placed in 100-mm-diameter Petri dishes lined with moistened filter paper and cultivated for 8 days. Subsequently, the seedlings were transferred to Yoshida nutrient solution. At the one-leaf and one-tiller stage (13 days after germination), drought stress was simulated by treating the seedlings with a 16% (w/v) PEG solution (prepared in Yoshida nutrient solution, pH 5.7–5.8) as the drought treatment group (Drought). Seedlings grown in Yoshida nutrient solution (pH 5.7–5.8) without PEG served as the control group (Control). The PEG treatment duration should be determined based on the specific experimental requirements of each result. The nutrient solution was replaced every 2 days. The cultivation environment was maintained at 22°C with a light intensity of 66 μmol·m⁻²·s⁻¹ and a photoperiod of 16-hour light/8-hour darkness.
- Were gas exchange measurements taken two days after the start of the water deficit? If so, why? Were any measurements conducted on the last day of water deficit? What light intensity was used in the LICOR during measurements, and at what time were they performed? Additionally, the authors mention “additional photosynthetic parameters were measured” (line 324) but do not specify which parameters or methods were used. The methodology section is to describe everything that was done and how.
Response:
We appreciate your valuable suggestion. We have revised the title. Yes, gas exchange measurements were conducted following a 2-day treatment with 16% PEG. The experimental design rationale was established based on observations from the gas exchange, the photosynthetic and DAB/NBT staining, which revealed a statistically significant divergence in chlorophyll content between drought-sensitive and drought-tolerant cultivars following 2-3 days of 16% PEG treatment. Consequently, all subsequent experiments were standardized using the hulless barley seedlings at the one-leaf-one-heart stage, subjected to 16% PEG-induced osmotic stress for 2-3 days to ensure experimental consistency.
All measurements in this study were conducted after 2-3 days of 16% PEG treatment, which strictly aligned with the final day of the water deficit period. Our experimental timeline was designed to capture critical physiological responses during the peak osmotic stress phase, with data collection therefore coinciding precisely with the endpoint of the drought simulation protocol.
Gas exchange measurements were conducted from 09:00 am to 4:00 pm on the hulless barley leaf after 2 days of 16% PEG treatment using a portable photosynthesis system coupled to a 3×3 cm2 leaf chamber (Li-6800; LICOR, Inc., Lincoln, NE, USA). The light intensity inside the leaf chamber was set to 100 µmol m-2 s-1, and the CO2 concentration surrounding the leaf was maintained at 400 µmol mol-1 with a CO2 mixer. The relative humidity (60%) and leaf temperature (22°C) were measured within an environment-controlled greenhouse. The aforementioned methodological details have been systematically incorporated into the revised manuscript. Line325
In response to reviewer feedback, the methodological details in line 324 have been rigorously supplemented with the following gas exchange parameters: Net photosynthetic rate (A); Transpiration rate (E); Stomatal conductance of water vapor (gsw); Total conductivity to water vapor (gtw); Intercellular CO2 concentration (Ci) and Stomatal conductance of CO2 (gtc). These parameters have been systematically integrated into the revised manuscript.Line329-341.
- How long were the plants subjected to dark adaptation before chlorophyll fluorescence analysis?
Response:
The chlorophyll fluorescence parameters were measured after 15 minutes of dark adaptation. Line347.
- What references were used for chlorophyll concentration and ROS analyses?
Response:
The total chlorophyll contents of individual plants were calculated as previously described (Wang Y, Zheng Y, Shi Y, Jiang D, Kuang Q, Ke X, Li M, Wang Y, Yue X, Lu Q, Hou X. YELLOW, SERRATED LEAF is essential for cotyledon vein patterning in Arabidopsis. Plant Physiol. 2024 Dec 2;196(4):2504-2516.). The ROS was performed as previously described (Yang X, Che Y, García VJ, Shen J, Zheng Y, Su Z, Zhu L, Luan S, Hou X. Cyclophilin 37 maintains electron transport via the cytochrome b6/f complex under high light in Arabidopsis. Plant Physiol. 2023 Aug 3;192(4):2803-2821.). These parameters have been systematically integrated into the revised manuscript. Line358.
- The data analysis section is missing. It is necessary to specify which statistical analyses were performed, which tests were applied, and which indices were used to determine significant differences. What was the experimental design?
Response:
Thank you for pointing this out. This was revised in the new version. Origin 2022 software was used for all statistical tests. Unpaired Student's t-test (2-tailed) was performed for the statistical analyses. These parameters have been systematically integrated into the revised manuscript. Line68-370.
The results indicate that drought-tolerant cultivars maintain physiological performance similar to control plants under stress, whereas drought-sensitive cultivars exhibit reduced performance. However, this is expected and does not provide novel insights. The discussion is weak and lacks integration with the findings. For example, why do the authors emphasize parameters such as gtw and gtc but fail to discuss them further? Additionally, no leaf water status parameters (e.g., relative water content, water potential) or other relevant metrics such as water use efficiency were measured, and in drought studies this data is of great importance. The reason for the lower NPQ in stressed plants is also not addressed.
Response:
Thank you sincerely for your comments. This study aims to construct a comprehensive evaluation system for drought resistance in hulless barley. By systematically screening key physiological and biochemical indicators of plant drought stress responses (including chlorophyll content, photosynthetic parameters, superoxide dismutase (SOD) activity, etc.), we will establish a multi-parameter integrated evaluation model for drought resistance classification. These findings will offer scientific support for drought-resistant breeding programs and cultivation management strategies specific to Tibetan hulless barley production systems.
Furthermore, in strict adherence to your recommendations, we have comprehensively revised the Discussion section and incorporated these methodological refinements into the updated manuscript version submitted for review.
Furthermore, the correct terminology is "fresh mass" and "dry mass," not "weight." Lastly, the measurement units of the analyzed parameters need to be reviewed.
Response:
Thank you for your suggestion. We have revised this accordingly.
Reviewer 2 Report
Comments and Suggestions for Authors
The study was devoted to screening water stress tolerance in seedlings of hulless barley genotypes. This common methodological approach does not allow to get a better insight into stress adaptation, however, some experimental and methodological results may be of interest to the reader. I also believe that this manuscript does not belong in IJ Molecular Sci. but rather to Plants or Agronomy.
The effect of PEG on the characteristics of barley (cereal) genotypes has been studied in hundreds of studies. Experiments with short-term stress induced by PEG (and other osmotically active substances) on juvenile plants often do not correlate with actual drought tolerance due to the many traits, mechanisms and factors involved in growth under field conditions. It is a great pity that the authors did not include in this study the monitoring of the effect of stress in sensitive and insensitive genotypes at later stages of development, in soil. The results obtained are thus of limited relevance for breeding.
The manuscript has many shortcomings which must be corrected.
The wording in the title "...markers for breeding climate-resilient crops" is exaggerated given the results from seedlings and the use of PEG
Abstract is weak, it requires significant rewriting, contains general claims (why does staple barley offer genetic? insight into stress adaption, genetic analysis was not preformed. What thresholds?) and almost no results. The claims are incomprehensible without knowledge of the methodology, for example, the reader does not know what the weak drought resistance of three uncharacterized cultivars is being compared to.
L.51 What chromium?
L56 or L291 Why do you refer to these different species when there are many studies on barley.
L72 Is there any indication why hulless barley should respond differently to water stress than other barley species?
Results
L82-91 Here, and elsewhere (L126), the text belongs rather to Introduction
L93 and L189 How the resistance and sensitivity in the varieties were determined, by field experiments? Were the results published?
Figures: The asterisks showing significant differences are not needed when the exact values of p are shown
L125 Unclear – “large range of variation and stability”, compare, for example, data of gsw and gtw
L156-157 Unclear - two different media?
L189 What you mean with “relatively”
L194 Did you averaged data for drought tolerant and sensitive varieties?
The Discussion demands significant improvement, it is often general, results are repeated not discussed, the experimental data and findings are little discussed and compared with relevant studies, especially, the different sensitivity of observed growth and physiological traits in tolerant and sensitive cultivars.
L232 The term “leaf wilting” was not used and quantified in Methods and Results
Materials and Methods
Add the description of statistical analysis
L334-335 Unclear
Author Response
The study was devoted to screening water stress tolerance in seedlings of hulless barley genotypes. This common methodological approach does not allow to get a better insight into stress adaptation, however, some experimental and methodological results may be of interest to the reader. I also believe that this manuscript does not belong in IJ Molecular Sci. but rather to Plants or Agronomy.
The effect of PEG on the characteristics of barley (cereal) genotypes has been studied in hundreds of studies. Experiments with short-term stress induced by PEG (and other osmotically active substances) on juvenile plants often do not correlate with actual drought tolerance due to the many traits, mechanisms and factors involved in growth under field conditions. It is a great pity that the authors did not include in this study the monitoring of the effect of stress in sensitive and insensitive genotypes at later stages of development, in soil. The results obtained are thus of limited relevance for breeding.
The manuscript has many shortcomings which must be corrected.
The wording in the title "...markers for breeding climate-resilient crops" is exaggerated given the results from seedlings and the use of PEG
Response:
Thank you for your suggestion. We have revised the title.
Abstract is weak, it requires significant rewriting, contains general claims (why does staple barley offer genetic? insight into stress adaption, genetic analysis was not preformed. What thresholds?) and almost no results. The claims are incomprehensible without knowledge of the methodology, for example, the reader does not know what the weak drought resistance of three uncharacterized cultivars is being compared to.
Response:
We appreciate your valuable suggestion. We have revised this accordingly.
L.51 What chromium?
Response:
We value your constructive feedback. Through systematic re-evaluation of the Introduction section, lines 50-51 were identified as being peripheral to the core objectives of this investigation and have consequently been removed to enhance conceptual precision and structural coherence. This refinement ensures stricter methodological alignment with our drought-response characterization framework.
L56 or L291 Why do you refer to these different species when there are many studies on barley.
Response:
Thank you for pointing this out. According to your suggestion, we have carefully reviewed the entire article and corrected this issue.
L72 Is there any indication why hulless barley should respond differently to water stress than other barley species?
Response:
Thank you for pointing this out. No comparative studies have been systematically conducted to date that conclusively demonstrate the differential responses of hulless barley and other barley to water stress under controlled experimental conditions.
Results
L82-91 Here, and elsewhere (L126), the text belongs rather to Introduction
Response:
We sincerely appreciate your insightful suggestions. In accordance with your recommendations, we have performed a rigorous critical evaluation of the entire introduction section. This analytical process has culminated in a substantially revised version that achieves enhanced logical coherence and establishes a more robust theoretical foundation.
L93 and L189 How the resistance and sensitivity in the varieties were determined, by field experiments? Were the results published?
Response:
The drought-resistant and drought-sensitive hulless barley varieties were selected based on established cultivation practices in Qinghai-Tibet Plateau combined with preliminary experimental data accumulated by our laboratory. These selection criteria were derived from systematic observations of crop performance under controlled drought simulations conducted during our previous research cycles.
Figures: The asterisks showing significant differences are not needed when the exact values of p are shown
Response:
We appreciate the importance of your valuable comment. In accordance with your suggestion, all asterisks indicating statistical significance have been consistently removed from the figures.
L125 Unclear – “large range of variation and stability”, compare, for example, data of gsw and gtw.
Response:
Compared to other parameters - including A (net photosynthetic rate), gtw (total water vapor conductivity), Ci (intercellular CO2 concentration), and gtc (CO2 stomatal conductance - the drought-sensitive varieties (YC85, YC88) exhibited the most significant variations in E (transpiration rate) and gsw (water vapor stomatal conductance) under drought treatment relative to the control group. In contrast, drought-tolerant varieties (ZY1252, ZY1100) demonstrated greater stability in maintaining both A and gsw values under identical stress conditions. This distinct combination of substantial variation in sensitive varieties and remarkable stability in tolerant varieties led us to select E and gsw as optimal drought resistance indices, as they collectively reflect both the dynamic response range and stress tolerance capacity. We have carefully reviewed the entire manuscript and corrected this issue.
L156-157 Unclear - two different media?
Response:
We sincerely thank you for raising this point. The term "two different media" corresponds to the control group and PEG treatment. The experimental materials were obtained from two different media(control and 16% PEG treatment). From each media, we collected the first fully unfolded leaf after 15 minutes of dark adaptation to measure chlorophyll fluorescence parameters.
L189 What you mean with “relatively”
Response:
We sincerely appreciate your thorough review. In response to your comments, we have revised the text to enhance clarity and precision. The specific modifications are as follows:
We selected three cultivars of hulless barley varieties for analysis: Drought-tolerant variety: ZY97; Drought-sensitive variety: YC83; Previously uncharacterized sensitive varieties: ZY673, ZY1403, and KL14. All selected varieties were subjected to experimental conditions with or without polyethylene glycol (PEG) treatment.
L194 Did you averaged data for drought tolerant and sensitive varieties?
Response:
Thank you for your question. For samples ZY673, ZY1403, and KL14, we employed the relative ratio method for calculation, with the specific formula: measured parameter values of the PEG-treated group / those of the control group.
The Discussion demands significant improvement, it is often general, results are repeated not discussed, the experimental data and findings are little discussed and compared with relevant studies, especially, the different sensitivity of observed growth and physiological traits in tolerant and sensitive cultivars.
Response:
Thank you for pointing this out. We have carefully reviewed the entire manuscript and corrected this issue.
L232 The term “leaf wilting” was not used and quantified in Methods and Results
Response:
We thank you for identifying this discrepancy. Since the 'leaf wilting' parameter was not quantified in our experiment, we have followed your recommendation by removing the relevant statement, which has enhanced the manuscript's logical flow.
Materials and Methods
Add the description of statistical analysis
Response:
Thank you for pointing this out. This was revised in the new version. Origin 2022 software was used for all statistical tests. Unpaired Student's t-test (2-tailed) was performed for the statistical analyses. These parameters have been systematically integrated into the revised manuscript.
L334-335 Unclear
Response:
We apologize for the confusion caused by our description. We have carefully reviewed the entire manuscript and corrected this issue. The chlorophyll fluorescence parameters for all experimental varieties were measured with 4 technical repetitions and 3 biological replicates.
Round 2
Reviewer 1 Report
Comments and Suggestions for Authors
The authors have substantially improved the manuscript, adhering to most of the suggestions; however, there is still a need for a review of terms, as there are inconsistencies between the terms used throughout the text (e.g.: Rfd), confusion in the names of the parameters, etc. In addition, the data obtained need to be related in the discussion; even though the new version is better, there is still a lack of greater depth in the evaluation and discussion of the data.
Author Response
The authors have substantially improved the manuscript, adhering to most of the suggestions; however, there is still a need for a review of terms, as there are inconsistencies between the terms used throughout the text (e.g.: Rfd), confusion in the names of the parameters, etc. In addition, the data obtained need to be related in the discussion; even though the new version is better, there is still a lack of greater depth in the evaluation and discussion of the data.
Response:
We appreciate your valuable feedback. We have reviewed the terminology throughout the text and implemented the necessary revisions. Additionally, the discussion section (lines 226-293) has been comprehensively rewritten to enhance clarity and strengthen the argumentation framework.
Reviewer 2 Report
Comments and Suggestions for Authors
The authors significantly modified, corrected, and edited the text, improving the quality of the manuscript.
I have only several minor comments
The authors wrote Response to my comment: “The drought-resistant and drought-sensitive hulless barley varieties were selected based on established cultivation practices in Qinghai-Tibet Plateau combined with preliminary experimental data accumulated by our laboratory. These selection criteria were derived from systematic observations of crop performance under controlled drought simulations conducted during our previous research cycles”, but they included only the part of the first sentence in the corrected manuscript. I suggest adding the whole text of the response to Materials and Methods (L306).
I do not understand the response to my comment: “L194 Did you averaged data for drought tolerant and sensitive varieties?”. The text remains unclear as to what these values, on lines 197 to 2002, represent. Why are you comparing the values of varieties with unknown sensitivity to the average of varieties with low and high sensitivity and separately?
Also, the response and modification of the text “….were measured with 4 technical repetitions and 3 biological replicates“ is not clear.
Author Response
The authors significantly modified, corrected, and edited the text, improving the quality of the manuscript.
I have only several minor comments.
The authors wrote Response to my comment: “The drought-resistant and drought-sensitive hulless barley varieties were selected based on established cultivation practices in Qinghai-Tibet Plateau combined with preliminary experimental data accumulated by our laboratory. These selection criteria were derived from systematic observations of crop performance under controlled drought simulations conducted during our previous research cycles”, but they included only the part of the first sentence in the corrected manuscript. I suggest adding the whole text of the response to Materials and Methods (L306).
Response:
Thank you for your suggestion. We have revised this accordingly. Line311-315
I do not understand the response to my comment: “L194 Did you averaged data for drought tolerant and sensitive varieties?”. The text remains unclear as to what these values, on lines 197 to 202, represent. Why are you comparing the values of varieties with unknown sensitivity to the average of varieties with low and high sensitivity and separately?
Response:
Thank you for pointing this out. We sincerely regret any misunderstanding that may have arisen. We are not comparing the values of cultivars with unknown sensitivity to the average of cultivars with low and high sensitivity and separately. We employed a systematic experimental design to comparatively analyze three cultivars with uncharacterized drought sensitivity under both drought-treated and control (CK) treatment. To avoid ambiguity, we have rewritten this section in line 190-205.
Also, the response and modification of the text “….were measured with 4 technical repetitions and 3 biological replicates” is not clear.
Response:
We appreciate the opportunity to clarify this methodological detail. Our experimental design included three independent biological experiments, all of which demonstrated complete consistency in their results. We have presented representative data from one biological replicate in the main figures to enhance visual clarity.
In accordance with your suggestion, we have now explicitly annotated the sample size (n) for each experimental group in the relevant figure legends:
Line 116/152/165/214: Mean ± SD (n=4)
Line 120/188/220: Mean ± SD (n=3)
Line 213: Mean ± SD (n=8)